# Targeted Next-Generation Sequencing for Comprehensive Testing for Selected Vector-Borne Pathogens in Canines

**DOI:** 10.3390/pathogens11090964

**Published:** 2022-08-24

**Authors:** Jobin J. Kattoor, Emma Nikolai, Barbara Qurollo, Rebecca P. Wilkes

**Affiliations:** 1Animal Disease Diagnostic Laboratory, Department of Comparative Pathobiology, College of Veterinary Medicine, Purdue University, West Lafayette, IN 47907, USA; 2Vector Borne Disease Diagnostic Laboratory, Department of Clinical Sciences, College of Veterinary Medicine, North Carolina State University, Raleigh, NC 27606, USA

**Keywords:** vector-borne pathogens, diagnostics, targeted next-generation sequencing

## Abstract

The standard for detecting vector-borne pathogens is real-time PCR (rtPCR). However, this requires many individual tests to obtain an accurate diagnosis. The purpose of this study was to develop and validate a targeted next-generation sequencing (NGS) assay for vector-borne pathogens. Pathogen target regions were amplified via PCR using two primer pools that were developed in conjunction with ThermoFisher Scientific, and barcoded DNA libraries were prepared and sequenced with the Ion Torrent S5 system. Data were assembled using SPAdes and mapped to a reference file containing sequences from the pathogens. The raw reads were analyzed to confirm the results. Test feasibility and analytical specificity were evaluated with type strains or validated positive clinical samples from dogs. The analytical sensitivity of the method was compared to Ct values obtained by rtPCR testing. Diagnostic sensitivity and specificity were assessed with a set of known positive and negative clinical samples based on rtPCR testing. Positive and negative percent agreements and Cohen’s kappa were calculated. The primer sets were specific for the intended targets, based on sequence analysis of the amplified products, and the method detected 17 different pathogens. Analytical sensitivity was equivalent to an rtPCR Ct value of approximately 35–36. The positive percent agreement was 92%, and the negative percent agreement was 88%. Cohen’s kappa was 0.804, which indicates almost perfect agreement between the rtPCR assays and the targeted NGS assay. Using a targeted method reduces the costs associated with NGS sequencing and allows for a 2–3 day turn-around time, making this a viable method for detection of vector-borne pathogens in canine whole blood samples.

## 1. Introduction

Vector-borne diseases are infectious diseases caused by a range of pathogens transmitted to animals and humans through a vector—a living organism, which in many cases is a blood-sucking arachnid (e.g., ticks, fleas, mosquitos). There has been a resurgence of vector-borne pathogens in recent years as a result of many contributing factors, including climate change, the deforestation of tropical forests, land-use change, urbanization, human population growth and migration, habitat fragmentation, animal movements, and biodiversity loss [1]. Animal movement, including the movement of animals across the country or from abroad for adoption or breeding, can contribute to the introduction of vectors or pathogens to new areas within the country, particularly if animals are moved from endemic areas because they may have subclinical infections [2,3]. Vector control and accurate diagnosis and treatment will remain the cornerstones of coping with vector-borne diseases in the foreseeable future [2]. Animal exports will continue, so veterinary professionals have a vital role to play as a first line for the detection of these pathogens. This is important not just from the standpoint of animal health but also for public health, as some of these pathogens are a cause of major zoonotic concern and constitute a serious human health hazard worldwide [3]. The diagnosis of a given zoonotic disease in a dog may guide subsequent testing and diagnosis of the same disease in a human patient [2].

Over the past few decades, the list of differential diagnoses of vector-borne diseases in dogs has increased substantially [2], which makes appropriate test selection a daunting task. Additionally, the similar and mainly nonspecific clinical signs associated with these diseases (fever, weight loss, inappetence, lethargy, or apathy) give no indication of the possible causative agent, adding to the challenge. Furthermore, co-infections with two or more pathogens are possible and may influence clinical signs and hematologic changes, thereby confounding the diagnosis [3]. Commonly used techniques for the diagnosis of these diseases are serology, direct detection by blood smear analysis, and molecular methods such as PCR (particularly real-time PCR). Culture is not used as a diagnostic technique because of the difficulties in isolating many of these organisms [4]. Blood smear analysis has low sensitivity compared to PCR and is unable to define the organism to the species level; serology is useful for the detection of exposure to these pathogens but is limited in its ability to confirm an active infection. For example, antibodies to *Rickettsia rickettsii* are often detected in clinically normal patients, particularly in regions with an intense tick population, largely due to past or recent infection with the other spotted fever group *Rickettsia* spp., some of which are non-pathogenic [3,4]. This type of cross-reactivity between pathogens is a limitation of serology, which can confuse diagnosis [3,4].

PCR assays are considered the most sensitive and specific tests for the detection of these pathogens in the acute stage of infection. However, a suite of molecular assays must be used in concert to make an accurate diagnosis [4]. Many molecular assays are commercially available as panels, allowing simultaneous detection of multiple vector-borne pathogens, including *Babesia* spp., *Hepatozoon* spp., and/or *Leishmania* spp., that may be present as co-infections but might otherwise be overlooked. Because protozoal infections require different treatments, recognizing these infections in co-infected patients is particularly important [4]. Many diagnostic laboratories do not routinely sequence amplicons produced from clinical samples, and although assays are validated against known rickettsial agents, the detection of unknown or unexpected organisms that do not cause disease in dogs is not readily achieved, and this may inadvertently result in a false-positive result [4]. 

The goal of this project was to develop a targeted next-generation assay for the detection of vector-borne pathogens. This type of assay allows for the detection of these pathogens in a single assay while also obtaining sequencing results to confirm a diagnosis. Targeted NGS has improved sensitivity and reduced costs compared to shotgun/deep sequencing, making this type of assay suitable for diagnostics [5,6]. 

## 2. Results

### 2.1. Feasibility and Analytical Specificity

The primer sets for 17/21 vector-borne pathogen species included in the panel were evaluated, and all successfully amplified the intended pathogens (Table 1). The assay was able to differentiate between the various species that were evaluated, including *Babesia canis* and *gibsoni; Ehrlichia canis* and *ewingii*; *Anaplasma phagocytophilum* and *platys*; and the haemotrophic *Mycoplasma* species included in the assay. While *Bartonella vinsonii* berkhoffii was not detected in the test samples, the primers for *B. vinsonii* did not detect *Bartonella henselae*. *Babesia canis* and *gibsoni* could be detected by the *Cytauxzoon felis* primers, but BLAST of the raw reads was able to differentiate between the two organisms. There was non-specific priming associated with multiple primer sets, but this could easily be discarded because the reads were too short (expected size, approximately 200 bp) and not the appropriate sequence. The validated strains/samples with significant amounts of pathogen could not be obtained for the other six pathogens. Additional primer sets included in the assay but not tested included: *Bartonella vinsonii berkhoffii, Yersinia pestis, Francesella tulerensis*, and additional Rickettsial species considered non-pathogenic for dogs. 

### 2.2. Analytical Sensitivity

The detection limit for the targeted NGS assay is similar to the real-time PCR Ct values in the mid-30s-approximately 35–36 (Table 2).

### 2.3. Diagnostic Sensitivity and Specificity

The percent positive and percent negative agreements, total agreement, and Cohen’s kappa were calculated in comparison to the real-time PCR results (Table 3). The methods had an almost perfect agreement for this sample set. Disparate results between the two methods are provided in Table 4.

## 3. Discussion

We developed a comprehensive method for the detection of infectious diseases in dogs, taking the guesswork out of determining which tests to use for diagnosis and potentially improving disease surveillance because of the comprehensive nature of the test. The method is a targeted next-generation sequencing (NGS) assay, which takes advantage of the amount of data that can be generated with NGS but also includes a PCR step prior to sequencing. This reduces the costs associated with the sequencing and provides adequate turn-around time for diagnostic use. False-negative results were obtained from two samples with Ct values above 35. This result was expected based on analytical testing that was performed. Interestingly, while *Babesia gibsoni* was missed in one sample by the targeted NGS panel, canine parvovirus type 2b was detected instead. The targeted panel includes primers for multiple canine pathogens (not just vector-borne pathogens). Unfortunately, there was no history available for this animal, so this is only speculation. This was a 7-year-old male castrated dog (unknown breed) from New Jersey. While parvovirus is not commonly diagnosed in 7-year-old dogs, it is possible. Another possibility is that the dog was recently vaccinated with a modified live virus, and we were detecting a viremia produced from the vaccine (some vaccines contain the CPV-2b strain). 

“False-positive” results were associated with the detection of *E. canis* in two samples that were considered negative by real-time PCR. While the sequences were consistent with *E. canis*, there were few reads detected in both samples. The low number of reads suggests little pathogen in these two samples, suggesting it was below the limit of detection by real-time PCR. An additional sample containing *E. canis* by real-time PCR (Ct 29.6) was not originally tested for haemotrophic *Mycoplasma* spp. because these were not part of the test panel. The targeted NGS panel detected *E. canis, Mycoplasma haemoparvum,* and *Candidatus Mycoplasma turicensis*. When the sample was submitted to a separate lab for testing by PCR and Sanger sequencing, the result provided was *Mycoplasma* spp., with *Mycoplasma haematoparvum* as the closest match. However, the sequence had multiple peaks suggestive of a co-infection. The targeted NGS was able to differentiate these two species in the sample, proving that the targeted assay is able to detect multiple pathogens, including different pathogens of the same genus, as well as speciate them in a single test. Based on the positive result by the separate lab, if we exclude this particular sample as a false positive result for the targeted NGS panel, it increases the NPA to 91.3%. 

Based on the almost perfect agreement between the targeted NGS assay and real-time PCR for the detection of these vector-borne pathogens, the use of this test for diagnostic purposes is acceptable. The targeted NGS panel is able to detect multiple pathogens in a single test, has a 2–3 day turn-around time, and costs approximately USD 200 per sample to perform. This includes the purchase of the primers for targeting that are available from ThermoFisher Scientific (primers are now IP and considered protected items of ThermFisher). This makes this method comparable to running multiple real-time PCR assays, but with the benefit of also obtaining the pathogen sequences to confirm the results. Loss of sensitivity compared to real-time PCR is a known limitation, and some pathogens present at very low amounts (Ct values in high 30 s) will be missed with this method. Unfortunately, vector-borne pathogens can commonly be found at very low levels in circulation [4]. However, some diagnostic laboratories consider samples with Ct values ≥ 35 to be suspect, and it is possible to miss pathogens at this very low level by real-time PCR as well. A blood sample collected at a single time point may not allow identification of infection by PCR or the targeted NGS method because of the fluctuation in the numbers of pathogens in circulation [4]. This is particularly true when tetracycline-based antibiotics are administered prior to the collection of the sample [4]. Thus, the slight reduction in analytical sensitivity compared to real-time PCR does not diminish the usefulness of this test. Targeted NGS is a sensitive and specific method to detect vector-borne pathogens in canine whole blood samples.

## 4. Materials and Methods

Primer Design. The primers were designed for multiple genes for each pathogen (including different species of each genus) based on specific and conserved regions in the literature. The design was developed in conjunction with ThermoFisher Scientific’s AgriSeq Bioinformatics team. Proprietary software was used to design primers with the same annealing temperature and incorporated into two separate pools in such a way as to avoid interactions between the primers themselves based on the design software. A FASTA file containing all the targeted sequences of each pathogen and an associated bed file was generated and uploaded to the S5 Torrent Server (ThermoFisher Scientific, Waltham, MA, USA). The FASTA Appendix A and bed Appendix A are included as Appendix A. The two primer pools designed for this assay are available from ThermoFisher Scientific (Custom AgriSeq T-GBS Vector-borne pathogens for Canine & Feline). Primers are IP and protected items of ThermoFisher.

Nucleic Acid Extraction. Convenience samples of EDTA-anticoagulated whole blood were submitted to the NC State-College of Veterinary Medicine, Vector-Borne Disease Diagnostic Laboratory (VBDDL) by veterinarians for vector-borne disease PCR testing. The DNA was extracted from 200 µL aliquots of whole blood using a QIAsymphony^®^ SP robot (QIAGEN, Valencia, CA, USA) and QIAsymphony^®^ DNA Mini Kit (QIAGEN, Valencia, CA, USA). Every extraction run included four aliquots of molecular-grade water as extraction controls, which were run as negative controls in all RT PCRs. 

Real-time PCRs. RT-PCRs primers and assay conditions used to detect vector-borne pathogens *Anaplasma*, *Babesia*, *Bartonella*, *Cytauxzoon, Ehrlichia*, *Hepatozoon*, hemotropic *Mycoplasma*, and *Rickettsia* have been previously described [7]. *Anaplasma*, *Babesia*, *Cytauxzoon felis*, and *Ehrlichia* RT-PCR positive samples were assayed again in a second, species-specific RT-PCR, and, if negative, the original genus-specific amplicon was sequenced [8]. All other amplicons from RT-PCR positive genera assays were sequenced. All of the amplicons were sequenced by Azenta Life Sciences (Raleigh, NC, USA) and the sequences aligned with the reference sequences from the NCBI GenBank. The extracted nucleic acids were shared with the Animal Disease Diagnostic Laboratory (ADDL) for targeted NGS. 

Targeted Next-Generation Sequencing Assay. A DNA library of each sample was prepared from the isolated nucleic acids from the canine whole blood samples (EDTA) using the Ion AmpliSeq™ Library Kit Plus (Thermo Fisher Scientific), and the barcoding of the samples was performed with Ion Xpress™ Barcode Adapters (Thermo Fisher Scientific). Briefly, 5 µL of 5X Ion AmpliSeq™ HiFi Mix from Ion AmpliSeq™ Library Kit Plus was mixed with 7.5 µL of extracted DNA. About 5 µL of each mixture was aliquoted into two PCR tubes to which 5 µL of the primer pools 1 and 2 were added, respectively. The PCR conditions applied were an initial denaturation at 99 °C for 2 min followed by 21 cycles of denaturation at 99 °C for 15 s and a combined annealing/extension step at 60 °C for 4 min. This was followed by the partial digestion of the amplicons, where each sample was treated with 2 µL of FuPa reagent and incubated at 50 °C for 10 min, 55 °C for 10 min, and 60 °C for 20 min. The amplified partially-digested amplicons were mixed with 2 µL of the adapter-barcode mixture, 4 µL of switch solution, and 2 µL of DNA ligase and incubated at 22 °C for 30 min, 68 °C for 5 min, and 72 °C for 5 min. The adapter-barcode ligated samples were purified using AMPure XP Reagent (Beckman Coulter, Brea, CA, USA) according to the manufacturer’s instructions. Purified libraries were quantified using a Qubit 4 fluorometer (Invitrogen, Thermo Fisher Scientific). The libraries were diluted to 100 pM concentration using nuclease-free water for loading an Ion Torrent™ Ion 510™ Chip using Ion 510™ and Ion 520™ and Ion 530™ Kit—Chef (Thermo Fisher Scientific), according to the manufacturer’s protocol. A positively loaded Ion 510™ Chip was sequenced in the Ion S5™ sequencing system (Thermo Fisher Scientific) according to the recommended protocol. The data were assembled using SPAdes (v5.12.0.0 on the Torrent Server) and mapped to a reference FASTA file containing sequences from the pathogens. Geneious Prime v. 2021.1.1 (https://www.geneious.com/prime/, accessed on 23 August 2022) was used to process the raw reads and the BLAST analysis (https://blast.ncbi.nlm.nih.gov/Blast.cgi, accessed on 23 August 2022) was performed to confirm the results.

Feasibility and Analytical Specificity. A set of validated positive clinical samples (canine, feline, or equine whole blood in EDTA tubes) and type strains (ATCC, Manassas, VA) were used to evaluate the primer pools for the ability to detect the pathogens and the specificity of the primer sets.

Analytical Sensitivity. A subset of the organisms was used to evaluate the sensitivity of the assay, including *Rickettsia rickettsii*, *Ehrlichia canis, Ehrlichia ewingii, Babesia gibsoni, Bartonella henselae, and Bartonella vinsonii berkhoffii.* Clinical samples used were tested by the VBDDL according to the described protocols, and the nucleic acids were shipped on ice to the ADDL. The samples had high Ct values (≥35), or if lower Ct values, then ten-fold serial dilutions were prepared from the nucleic acids to dilute the amount of pathogen by approximately 3.3 Ct values per 10-fold dilution. The targeted NGS panel was used to evaluate each dilution that provided a Ct value above 30.

Diagnostic Sensitivity and Specificity. A separate set of clinical samples was used to evaluate diagnostic sensitivity and specificity. These were nucleic acids either provided by VBDDL, which were tested as previously described, or from a laboratory at Texas A&M that had been tested using validated methods. The results from these other labs (real-time PCR) were compared with results from the targeted NGS assay performed at the ADDL. For samples to be considered true positives, all results obtained from the targeted NGS panel were required to match results obtained by real-time PCR. If additional organisms were detected by NGS, these samples were considered a false positive. The percent positive and percent negative agreements, total agreement, and Cohen’s kappa were calculated. 

## Figures and Tables

**Table 1 pathogens-11-00964-t001:** Feasibility and Analytical Specificity Testing.

Organism	Source
*Anaplasma phagocytophilum*	Validated clinical sample, Ct 20 ^a^
*Anaplasma platys*	Validated clinical sample, Ct 28 ^a^
*Ehrlichia canis*	Validated clinical sample, Ct 24.3 ^a^
*Ehrlichia ewingii*	Validated clinical sample, Ct 34.42 ^a^
*Babesia canis*	Synthetic DNA^b^
*Babesia gibsoni*	Validated clinical sample, Ct 18.05 ^a^
*Bartonella henselae*	Strain Houston-1 ^b^
*Leishmania infantum*	Strain LIVT-2 ^b^
*Cytauxzoon felis*	Validated clinical sample, 22.9 ^a^
*Borrelia burgdorferi*	Strain B31 ^b^
*Mycoplasma haemocanis*	Validated clinical sample, 21.5 ^a^
*Mycoplasma haematoparvum*	Validated clinical sample, 23.6 ^a^
*Mycoplasma haemofelis*	Validated clinical sample, 24.3 ^a^
*Mycoplasma haemominutum*	Validated clinical sample, 21.2 ^a^
*Rickettsia rickettsii*	Validated clinical sample, Ct 35.1 ^a^
Eastern equine encephalitis	Validated clinical sample, Ct 27 ^c^
West Nile virus	Validated clinical sample, Ct 26 ^c^

^a^ Vector Borne Disease Diagnostic Laboratory, North Carolina State University; ^b^ ATCC; ^c^ Animal Disease Diagnostic Laboratory, Purdue University.

**Table 2 pathogens-11-00964-t002:** Analytical Sensitivity Testing.

Organism	Positive/Negative Result Obtained for Targeted NGS Assay Comared to Ct Value Obtained by PCR Assays
*Rickettsia rickettsii*	Positive at 35, negative at 38
*Ehrlichia canis*	Positive at 36
*Ehrlichia ewingii*	Positive at 35 and 36, negative at 37
*Babesia gibsoni*	Positive at 36
*Bartonella henselae*	Positive at 33, negative at 36
*Bartonella vinsonii berkhoffii*	Negative at 36 *

* Did not have samples with lower Ct values to evaluate.

**Table 3 pathogens-11-00964-t003:** 2 × 2 Table displaying comparative results for 51 clinical samples obtained from dogs.

	Comparative Method (qPCR)	
**Candidate Method (Targeted NGS Panel)**	**Positive**	**Negative**
Positive	24	3
Negative	2	22

Positive Percent Agreement (95% CI)—92.31% (74.87% to 99.05%). Negative Percent Agreement (95 CI)—88.00% (68.78% to 97.45%). Accuracy (total percent agreement with 95% CI) 90.20% (78.59% to 96.74%). Cohen’s Kappa-0.804, Almost perfect agreement.

**Table 4 pathogens-11-00964-t004:** Disparate results obtained between real-time PCR assays and targeted NGS assay.

Organism	qPCR Ct Value	Targeted NGS
*Babesia gibsoni*	37.18, 40.0, and 35.82 *	Parvovirus
*Ehrlichia canis*	37	Negative
*Ehrlichia canis*	Negative	*E. canis*^#^ 69 reads
*Ehrlichia canis*	Negative	*E. canis*^#^ 169 reads
*Mycoplasma haemoparvum, and Candidatus M. turicensis*	Not included in the test panel	*Mycoplasma haemoparvum* and *Candidatus M. turicensis*

* Tested with 3 different primer sets by qPCR; ^#^ Total number of reads detected from 2 primer sets.

## Data Availability

Not applicable.

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
