# Peer review of "Targeted Next-Generation Sequencing for Comprehensive Testing for Selected Vector-Borne Pathogens in Canines"

_pathogens, 2022, doi:10.3390/pathogens11090964_

Round 1

Reviewer 1 Report

Jobin J. Kattoor et al. described the development and validation of a targeted next-generation sequencing (NGS) assay for twenty-one vector-borne pathogens, consisting of almost protozoas, in diagnosis of a given zoonotic disease in a dogPathogen target regions were amplified via multiplex PCR, and barcoded DNA libraries were sequenced with the Ion Torrent S5 systemAnalytical sensitivity of the method was compared to rtPCR testing. Diagnostic sensitivity and specificity were assessed with a set of known positive and negative clinical samples.

The primary issue with this manuscript is the simplistic nature of the study. Targeted, even multiplex PCR with NGS is not novel in itself now. The challenge is that the study has limited complexity in terms of study design or analyses, so as not be enough for article

Below are other comments for authors considerations:
1. Please provide more details about the multiplex PCRHow many pools, how the primers were divided in these pools and the sequences for the primers.

2. Please explain how the method can deal with the situation of co-infections with two or more pathogens.

3. In the manuscript, vector-borne pathogens mentioned in the manuscript consisted mainly of protozoas, so protozoal pathogens may be more appropriate.

4. In line 86, what does the 17/21 mean?

5. Line 176, the primers should be provided.

Author Response

The primary issue with this manuscript is the simplistic nature of the study. Targeted, even multiplex PCR with NGS is not novel in itself now. The challenge is that the study has limited complexity in terms of study design or analyses, so as not be enough for “article”

We agree that targeted NGS is no longer a novel concept. However, as far as pathogen detection goes, particularly for a clinical diagnostic lab, targeted NGS is still in its infancy. Most of the work is geared towards metagenomics, which lacks sensitivity (and potentially specificity). We have validated a new method that provides adequate sensitivity and specificity for use in a diagnostic laboratory, which is held to higher standards with regard to assay design than research labs. Standard analyses required for diagnostic test validation were performed. 

Below are other comments for authors considerations:
1. Please provide more details about the multiplex PCR. How many pools, how the primers were divided in these pools and the sequences for the primers.

Much of this is described in lines 170-179. There are two primer pools. The proprietary software from ThermoFisher designates the pools. I provided sequence regions known to be useful for detection of these pathogens and then evaluated all the designed primers in silico for specificity. Ones that were not specific for the intended pathogen were removed. The primer sequences are considered proprietary, but these pools are available from ThermoFisher so the work can be reproduced in another laboratory.

  1. Please explain how the method can deal with the situation of co-infections with two or more pathogens. This is based on the targeting using the multiplex assay. Many of the PCR primers in the two pools are specific for a single organism. There are some that are genera specific (not species specific). The fasta file used for data analysis has sequences for the various species. Sequences generated in the assay are compared to the file to ID specific species present. We also blast the sequences to further confirm identity. Due to the targeting, we can amplify all pathogens present for which we have primers, significantly increasing our ability to detect these pathogens, whether present alone or in combination.
  2. In the manuscript, “vector-borne pathogens”mentioned in the manuscript consisted mainly of protozoas, so protozoal pathogens may be more appropriate. Actually, most of the pathogens are bacterial, not protozoal. However, they are all vector-borne pathogens, so we request maintaining the use of this descriptor.
  3. In line 86, what does the “17/21”mean? As mentioned at the end of the paragraph- "Additional primer sets included in the assay but not tested included: Bartonella vinsonii berkhoffii, Yersinia pestis, Francesella tulerensis, and additional Rickettsial species considered non-pathogenic for dogs." We could not obtain a sample with enough B. vinsonii for us to detect. Y. pestis and F. tulerensis are select agents, and we are not a select agent lab, so we did not try to test these. However, we considered these important for inclusion in the assay. 
  4. Line 176, the primers should be provided. Unfortunately we cannot share the primer sequences. This assay was developed in conjunction with the bioinformatics team at ThermoFisher and they consider these sequences proprietary. However, as mentioned, the primer pools are available. If ThermoFisher changes it mind, I will provide the sequences at a later date as supplementary material.

Author Response

Point 1: This developed assay requires sophisticated equipment to do NGS than other diagnostic methods such as LAMP and multiplex PCR. Furthermore, time required to get the results is longer than other methods. Cost is also another factor as you have indicated in your discussion that, “The targeted NGS panel is able to detect multiple pathogens in a single test, has a 2-3 day turn-around time, and costs approximately $200 per sample to perform”. How applicable is this assays in disease surveillance because we need rapid, cost effective and applicable in resource limited settings such as developing countries where the disease burden is high?

While it is true that tests are needed for resource limited settings, this assay was not touted as filling that need. The targeting can be applied to a different sequencing platform, such as Oxford Nanopore technology, which is much less expensive and can be used in the field. We still need additional comprehensive testing for dogs in the US, due to expanding range of vectors, new vector introductions, etc. This test meets that need in a population that also can serve as sentinels for human infections. The time to result is only extended by 1-2 days, but sequences are also obtained, which is not possible with regular PCR. This sequencing information is useful for some of the genera in this assay. For example- not all the Rickettsia species are pathogenic.

  1. The title is somewhat misleading as it may imply that this assay can detect all vector-borne pathogens of dogs which is not the case. I would propose to put the word “selected” before the word vector-borne.

Changed as suggested

2.    In line 33, ticks are not insects they are arachnids. Arthropods can be a better general term than insects.

Changed as suggested
3.    Sentences from lines 37-40 and lines 40-44 seems a repetition because they are describing the movement of animals as possible means of transmission of pathogens. The information can be summarized and put in one sentence.

Changed to: "Animal movement, including movement of animals across the country or from abroad for adoption or breeding, can contribute to introduction of vectors or pathogens to new areas within the country, particularly if animals are moved from endemic areas because they may have subclinical infections." 
4.    Line 47, the spelling of health should be corrected.

Corrected
5.    Line 52, daunting should be changed to “a daunting task”.

Changed as suggested
6.    In line 56, what does “laboratory changes” mean and how does it affect diagnosis of vector-borne pathogens?

Changed to hematologic changes to be more specific. Some of these pathogens are not pathogenic in an otherwise healthy dog unless there is a coinfection (eg. B. canis). So, for example, if B. canis was detected but the coinfection was missed, this would be a problem.
7.    In line 60, the statement that blood smear lacks sensitivity and specificity is incorrect. Although they have low sensitivity, the specificity is very high as a samples that is positive will also be positive on the other tests such as PCR or serology. However, they cannot be used to identify pathogens to species level.

Changed to: "Blood smear analysis has low sensitivity compared to PCR and is unable to define the organism to the species level; serology is useful for detection of exposure to these pathogens but is limited in its ability to confirm an active infection." 
8.    In line 62, R rickettsia should be written in full as the species is being mentioned for the first time. This should also be in italics.

Corrected

9.    In line 64, Rickettsia should be in italics

Corrected
10.    In line 67, PCR assays are more specific but not sensitive than serology. Please check and revise accordingly.

Limited to acute infections for accuracy.

11.    Please all species names should be written in full when appearing for the first time and abbreviate later.

Corrected
12.    In Table 2 on Analytical Sensitivity Testing why was this done on few pathogens not all the 19 investigated in this study and why some pathogens have negative Ct values missing?

Honestly this was due to funding. We did what we could with the 100K grant (which also included funds for post-doc help). We chose some organisms we more commonly see in dogs in the tested regions and also evaluated bacteria and a parasite. We got similar results among these and expect the results would be similar for the other organisms in the panel as well.

The title of the second column of table 2 was changed. The pos/neg refers to the result for the targeted NGS panel, compared to the Ct values obtained for these samples by PCR assays.

Round 2

Reviewer 2 Report

The authors have addressed the concerns raised and the manuscript can be accepted.

Author Response

Thank you!